# Breaking Habits: On the Role of the Advantage Function in Learning Causal State Representations

## Abstract

Recent work has shown that reinforcement learning agents can develop policies that exploit spurious correlations between rewards and observations. This phenomenon, known as policy confounding, arises because the agent's policy influences both past and future observation variables, creating a feedback loop that can hinder the agent's ability to generalize beyond its usual trajectories. In this paper, we show that the advantage function, commonly used in policy gradient methods, not only reduces the variance of gradient estimates but also mitigates the effects of policy confounding. By adjusting action values relative to the state representation, the advantage function downweights state-action pairs that are more likely under the current policy, breaking spurious correlations and encouraging the agent to focus on causal factors. We provide both analytical and empirical evidence demonstrating that training with the advantage function leads to improved out-of-trajectory performance.

## 1 Introduction

Imagine a robot trained to perform two tasks: first, navigate from your office to the coffee machine, retrieve a cup of coffee, and return; second, go to the printer room, make copies, and return. There are two possible routes to the coffee machine: one through the printer room and another through a direct corridor. Since the corridor route is shorter, the robot typically avoids the printer room when fetching coffee. However, one day, the corridor is blocked and the robot takes the path through the printer room, unexpectedly returning with a copy of a paper titled *Breaking Habits*.

Why did this happen? After repeatedly performing these tasks, the robot incorrectly associated the printer room with the need to make copies, turning this misassociation into a habit. While this habit works fine under normal conditions, it fails when the robot is forced to deviate from its usual path. This failure mode, referred to as *out-of-trajectory generalization*, was explored by Suau et al. (2024) in the context of reinforcement learning (RL). The authors showed that such issues arise because the agent's policy introduces spurious correlations (Pearl et al., 2016) between rewards and observations, a phenomenon they termed *policy confounding*.

**Contributions**  In this paper, we observe that the advantage function, commonly used in many policy gradient methods, not only reduces the variance of gradient updates but also plays a crucial role in mitigating this issue. We demonstrate, both analytically and experimentally, that using the advantage function encourages the agent to learn representations that rely more on the true causal factors. It achieves this by scaling state-action pairs according to their probability under a given policy and state representation, effectively breaking the spurious correlations introduced by the agent's policy and enabling improved out-of-trajectory generalization.

To support the theoretical findings, experiments are conducted in three simple environments. The results indicate that agents trained with $Q$-values fail to generalize beyond their usual trajectories, whereas agents trained with the advantage function exhibit robustness to trajectory deviations. Furthermore, an analysis of the learned state representations reveals that the latter focus on causal factors rather than exploiting spurious correlations.

Finally, we examine the impact of implementation choices, such as batch size and advantage normalization. Through experiments, we show that these factors can significantly affect the learned state representations.

## 2 Example: Key2Door

We use the Key2Door environment throughout the paper to illustrate the ideas. Figure 1 depicts a simple gridworld. The agent's objective is to collect a key at the beginning of a corridor and then open the door at the end.

The agent's state is described by two variables: its current location $L \in \{1, 2, \dots, 6\}$, and a binary variable $X \in \{0, 1\}$ indicating whether the agent has collected the key. At each location, the agent can move left $(A = 0)$ or right $(A = 1)$. When the agent reaches the door, the episode terminates and it receives a reward of $+1$ if it has collected the key, or 0 otherwise. The agent also receives a small per-timestep penalty of $-0.01$ to encourage shorter paths.

During training, the agent always starts at location 2. As a result, once the agent has collected the key and is moving toward the door, it can disregard the key when it reaches location 3. The agent can learn to use its location as a proxy for whether it has the key. This approach works well only if the agent follows the optimal policy (purple arrow); under random or suboptimal policies, location alone does not reliably indicate key possession.

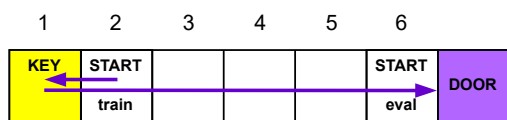

Figure 1: Key2Door environment.

This environment highlights the risk of policy confounding. An agent that relies solely on its location will fail when evaluated in the same task, but starting, for example, from location 6. The agent will attempt to move right to open the door immediately after the episode starts, even though it has not collected the key.

## 3 Related work

The term policy confounding was introduced by Suau et al. (2024) to describe a phenomenon in which policies, by influencing both past and future observation variables, induce spurious correlations. These correlations can lead the agent to develop shortcuts, referred to as habits, that are effective only within the trajectories the agent typically follows but fail to generalize out-of-trajectory. Out-of-trajectory (OOT) generalization is a specific instance of the broader problem of out-of-distribution (OOD) generalization (Kirk et al., 2023). Unlike OOD generalization, which focuses on adapting to environments with different rewards (Taylor & Parr, 2009), observations (Mandlekar et al., 2017; Zhang et al., 2020a), or transition dynamics (Higgins et al., 2017), OOT generalization seeks to enable agents to generalize across alternative trajectories within the same environment. Empirical evidence of policy confounding has been reported in several studies (Machado et al., 2018; Song et al., 2020; Lan et al., 2023; He et al., 2024; Weltevrede et al., 2024). Similarly, prior work has examined confounding in the context of imitation learning (De Haan et al., 2019; Zhang et al., 2020b; Tennenholtz et al., 2021; Ding et al., 2023). However, unlike policy confounding, these works focus on cases where the agent does not actively contribute to the formation of spurious correlations.

The connection between the advantage function and causality has been explored in prior work. Corcoll & Vicente (2022) use the advantage function to differentiate between agent-driven effects and external environmental factors, constructing a hierarchy of transformations that the agent can perform on its environment. Pan et al. (2022) argue that the advantage function can be interpreted as a measure of an action's causal effect on return and introduce a method for learning advantages directly from on-policy data without relying on a value function. This approach was later extended to support off-policy data (Pan & Schölkopf, 2024). While not explicitly framing the advantage function in causal terms, Raileanu & Fergus (2021) describe it as a measure of the expected additional return gained by selecting a particular action over following the

current policy. Their experiments suggest that the advantage function is less prone to overfitting to certain environment-specific idiosyncrasies.

Our work is strongly influenced by Chung et al. (2021), who showed that the choice of baseline in policy gradient updates can sometimes lead to overly aggressive updates, causing what the authors term committal behavior, which may result in suboptimal policies. Mei et al. (2022) further analyzed the theoretical properties of state-value baselines (i.e., advantages), demonstrating that they moderate update aggressiveness and ensure convergence to the optimal policy. As we discuss in the following sections, these insights are closely connected to the findings presented in this work.

## 4 Preliminaries

In this section, we define the notation, introduce the problem formulation, and review the necessary background, including the notions of state representation and policy confounding. For a more comprehensive discussion of these concepts, we refer the reader to Suau et al. (2024).

### 4.1 Notation

Random variables are denoted by capital letters (e.g., $S$), their values by lowercase letters (e.g., $s$), and their domains by calligraphic letters (e.g., $\mathcal{S}$). For a set of random variables $F = \{F^1, \ldots, F^{|F|}\}$ with corresponding domains $\mathcal{F}^1, \ldots, \mathcal{F}^{|F|}$, we use $\times \mathcal{F} = \mathcal{F}^1 \times \cdots \times \mathcal{F}^{|F|}$. This notation represents all possible combinations of values for the variables in $F$. A specific assignment of values to the variables in $F$ is denoted by $\langle f^1, \ldots, f^{|F|} \rangle \in \times \mathcal{F}$.

### 4.2 Problem Formulation

We formalize the decision-making problem using the standard framework of Markov decision processes.

**Definition 1** (MDP). A Markov decision process (MDP) is a tuple $\langle \mathcal{S}, \mathcal{A}, T, R \rangle$, where $\mathcal{S}$ is the set of states, $\mathcal{A}$ is the set of actions, $T : \mathcal{S} \times \mathcal{A} \to \Delta(\mathcal{S})$ is the transition function, and $R : \mathcal{S} \times \mathcal{A} \to \mathbb{R}$ is the reward function.

In this work, we focus on a subclass of MDPs in which states are represented by a set of observation variables, or factors (Boutilier et al., 1999). This representation is common in tasks that involve modeling policies and value functions using function approximators (François-Lavet et al., 2018). These observation variables typically describe features of the agent's state in the environment.

**Definition 2** (FMDP). A Factored Markov decision process (FMDP) is an MDP in which the state space is defined by a set of state variables, $F = \{F^1, \ldots, F^{|F|}\}$ . Each variable $F^i$ takes values its domain $\mathcal{F}^i$. Consequently, each state $s \in \mathcal{S}$ corresponds to a unique combination of values for these variables, $s = \langle f^1, \ldots, f^{|F|} \rangle \in \times \mathcal{F} = \mathcal{S}$.

### 4.3 State representations

The agent's objective is to find a policy $\pi : \mathcal{S} \to \Delta(\mathcal{A}) \in \Pi$ that maximizes the expected discounted sum of rewards (Sutton & Barto, 2018). However, learning a policy that conditions on every observation variable might be impractical, especially in scenarios with a large number of variables. Fortunately, in many problems, not all variables are essential, and compact state representations can be found that are sufficient for solving the task at hand (McCallum, 1995). This is where function approximators, such as neural networks, come into play. When used to model policies and value functions, they may learn to ignore certain state variables if they are deemed unnecessary for predicting rewards or transitions.

**Definition 3** (State representation). A state representation is a function $\Phi : \mathcal{S} \to \bar{\mathcal{S}}$, where $\mathcal{S} = \times \mathcal{F}$, $\bar{\mathcal{S}} = \times \bar{\mathcal{F}}$, and $\bar{F} \subseteq F$.

Intuitively, a state representation $\Phi(s_t)$ is a projection of a state $s \in \mathcal{S}$ onto a lower-dimensional space $\bar{S}$, defined by a subset of its variables. We use $\{s\}^{\Phi} = \{s' \in \mathcal{S} : \Phi(s') = \Phi(s)\}$ to denote the equivalence class of $s$ under $\Phi$.

Throughout this paper, we use $\Phi$ as a notational device to reason about the information retained by function approximators. Rather than assuming an explicit abstraction step, $\Phi$ represents the effective state information on which a learned policy or value function conditions. In this sense, $\Phi$ captures the implicit state compression induced by function approximation, allowing us to analyze which variables are ignored and the consequences of doing so.

In the Key2Door example a potential state representation could be $\Phi(s_t) = \langle l_t \rangle$ for all $s_t \in \mathcal{S}$. This representation retains only the agent's current location, ignoring all other variables. Therefore, all states that share the same location belong to the same equivalence class.

## 4.4 Reward and Transition Function under a State Representation

When multiple ground states map to the same state representation, rewards and transition dynamics must be defined by aggregating over those states.

**Definition 4** (Policy-induced reward and transition functions under $\Phi$)**.** Let $\Phi : \mathcal{S} \to \bar{\mathcal{S}}$ be a state representation and let $\pi$ be a policy inducing a stationary distribution $d^\pi$ over $\mathcal{S}$. The *policy-induced reward and transition function* under $\Phi$ are defined as

$$R_\Phi^\pi(\Phi(s_t), a_t) := \sum_{s_t' \in \{s_t\}^\Phi} P^\pi(s_t' \mid \Phi(s_t)) \, R(s_t', a_t),$$

$$T_\Phi^\pi(\Phi(s_{t+1}) \mid \Phi(s_t), a_t) := \sum_{s_{t+1}' \in \{s_{t+1}\}^\Phi} \sum_{s_t' \in \{s_t\}^\Phi} P^\pi(s_t' \mid \Phi(s_t)) \, T(s_{t+1}' \mid s_t', a_t),$$

where $\{s\}^\Phi := \{s' \in \mathcal{S} : \Phi(s') = \Phi(s)\}$ and

$$P^\pi(s_t' \mid \Phi(s_t)) := \frac{d^\pi(s_t')}{\sum_{s \in \{s_t\}^\Phi} d^\pi(s)}.$$

## 4.5 Markov State Representations

Not all state representations are sufficient for solving the task. Some representations may exclude variables that carry valuable information. For example, in the Key2Door example, knowing whether the key has been collected is essential for selecting optimal actions.

**Definition 5** (Markov state representation)**.** A state representation $\Phi$ is *Markov* if, for all $s_t, s_{t+1} \in \mathcal{S}$, $a_t \in \mathcal{A}$, and $\pi \in \Pi$,

$$R(s_t, a_t) = R_\Phi^\pi(\Phi(s_t), a_t) \quad \text{and} \quad \sum_{s_{t+1}' \in \{s_{t+1}\}^\Phi} T(s_{t+1}' \mid s_t, a_t) = T_\Phi^\pi(\Phi(s_{t+1}) \mid \Phi(s_t), a_t).$$

Note that although $R_\Phi^\pi$ and $T_\Phi^\pi$ are defined with respect to a policy $\pi$, Markov state representations require the above equalities to hold for all policies.

The above definition is analogous to the notion of bisimulation (Dean & Givan, 1997; Givan et al., 2003) or model-irrelevance state abstraction (Li et al., 2006). Markov state representations are guaranteed to be behaviorally equivalent to the original representation.

However, an agent following a fixed policy $\pi$ may admit even more compact representations that are Markov only over the states and actions encountered under that policy.

**Definition 6** ($\pi$-Markov state representation)**.** A state representation is $\pi$-*Markov*, denoted $\Phi^\pi$, if, for all states visited under $\pi$, $s_t, s_{t+1} \in \mathcal{S}^\pi$ and all $a_t \in \text{supp}(\pi(\cdot \mid s_t))$,

$$R(s_t, a_t) = R_\Phi^\pi(\Phi^\pi(s_t), a_t) \quad \text{and} \quad \sum_{s_{t+1}' \in \{s_{t+1}\}^{\Phi^\pi}} T(s_{t+1}' \mid s_t, a_t) = T_\Phi^\pi(\Phi^\pi(s_{t+1}) \mid \Phi^\pi(s_t), a_t),$$

### 4.6 Policy Confounding

The phenomenon of *policy confounding* plays a critical role in the emergence of simplified state representations. When learning from data generated by a fixed policy, a function approximator may exploit spurious correlations induced by that policy, rather than relying on the true causal factors governing rewards and transitions. As a result, the learned representation may appear sufficient under the policy being followed, while failing to generalize to interventions or policy changes.

To formalize this notion, we adopt a causal perspective and use the do-operator (Pearl et al., 2016), which represents physical interventions that break policy-induced correlations. In our setting, $\mathrm{do}(\Phi(s_t))$ corresponds to intervening on the variables selected by the state representation $\Phi$, setting them to a particular value while allowing the remaining state variables to vary according to the environment's transition dynamics, independently of the policy being followed.

**Definition 7** (Policy confounding). A $\pi$-Markov state representation $\Phi^\pi$ is said to be *confounded by policy* $\pi$ if there exist $s_t, s_{t+1} \in \mathcal{S}$ and $a_t \in \mathcal{A}$ such that

$$R_\Phi^\pi(\Phi^\pi(s_t), a_t) \;\neq\; R_\Phi^\pi(\mathrm{do}(\Phi^\pi(s_t)), a_t),$$

or

$$T_\Phi^\pi(\Phi^\pi(s_{t+1}) \mid \Phi^\pi(s_t), a_t) \;\neq\; T_\Phi^\pi(\Phi^\pi(s_{t+1}) \mid \mathrm{do}(\Phi^\pi(s_t)), a_t).$$

Intuitively, a representation is policy-confounded if the reward or transition dynamics depend on correlations induced by the policy, rather than on the underlying causal factors. Under intervention, these correlations are broken, revealing discrepancies in the induced rewards or transitions.

For example, in the Key2Door environment, when following the optimal policy $\pi^*$ (purple path in Figure 1), being at location $L = 6$ perfectly predicts that the agent has the key. Consequently, $R_\Phi^{\pi^*}(L = 6) = +1$. However, under the intervention $\mathrm{do}(L = 6)$, the agent may or may not have collected the key, yielding $R_\Phi^{\pi^*}(\mathrm{do}(L = 6)) \in \{-1, +1\}$.

Strictly speaking, the only representations that are guaranteed to be invariant to such interventions are Markov state representations (Suau et al., 2021, Theorem 1). These representations are independent of the agent's policy and necessarily include all causal factors governing rewards and transitions. In the remainder of the paper, we therefore refer to Markov state representations as *causal state representations*.

### 4.7 Policy gradient and advantage function

Suau et al. (2024) demonstrated that the phenomenon of policy confounding is particularly problematic when training agents with on-policy methods. This is because, when updating the policy, on-policy methods rely solely on trajectories collected using the current policy. This contrasts with off-policy methods, where the agent is trained on trajectories generated by multiple policies, thus broadening the trajectory distribution and reducing the risk of the agent picking up on spurious correlations present in specific trajectories.

A popular family of on-policy methods includes those that directly optimize the policy by following the gradient of the expected return with respect to the policy parameters, $\theta$. The policy gradient theorem (Marbach & Tsitsiklis, 1999; Sutton et al., 1999) formalizes this as:

$$\nabla_\theta J(\pi_\theta) = \mathbb{E}_{s_t, a_t \sim \pi_\theta} \left[ \frac{\nabla_\theta \pi_\theta(a_t \mid s_t)}{\pi_\theta(a_t \mid s_t)} Q^\pi(s_t, a_t) \right], \tag{1}$$

where $Q^\pi(s_t, a_t)$ is the action-value function under policy $\pi$.

Following this gradient increases the likelihood of sampling actions that lead to high returns while reducing the probability of actions leading to lower returns. However, in practice, computing the exact gradient is infeasible. Instead, we approximate the gradient using sample estimates from trajectories collected by the policy, which introduces high variance.

A common strategy to reduce variance is to subtract the state value function $V^\pi(s_t)$ from the $Q$-value (Baird, 1994; Greensmith et al., 2001), leading to the definition of the advantage function:

$$A^\pi(s_t, a_t) := Q^\pi(s_t, a_t) - V^\pi(s_t). \tag{2}$$

This adjustment does not introduce bias, as the value function is independent of the action, but significantly reduces the variance of the gradient estimation.

## 5 The role of the advantage in learning causal state representations

In this section, we show that beyond its well-known role in variance reduction, the advantage function implicitly counteracts policy confounding by reweighting state–action samples within representation-induced equivalence classes. This reweighting attenuates spurious correlations introduced by the policy, thus preventing the agent from forming habits and biasing learning toward variables that are causally relevant for rewards and transitions. All proofs for the theoretical results in this section can be found in Appendix A.

### 5.1 Value and advantage function under a state representation

Computing the advantage typically involves fitting a model to approximate the value function. Conceptually, the value function can be decomposed into a state representation function $\Phi(s)$, which projects the states into a lower-dimensional space defined by the subset of variables, and a function $V^\pi(\Phi(s))$ that maps the state representation to a scalar value.

**Definition 8** ($V^\pi$ and $Q^\pi$ under $\Phi$). Let $\Phi : \mathcal{S} \to \bar{\mathcal{S}}$ be a state representation (Definition 3) that induces an equivalence class $\{s\}^\Phi := \{s' \in \mathcal{S} : \Phi(s') = \Phi(s)\}$. Then the state value $V_\Phi^\pi : \bar{\mathcal{S}} \to \mathbb{R}$ and state-action value $Q_\Phi^\pi : \bar{\mathcal{S}} \times \mathcal{A} \to \mathbb{R}$ under $\Phi$ are defined as:

$$V_\Phi^\pi(\Phi(s_t)) := \sum_{s'_t \in \{s_t\}^\Phi} P^\pi(s'_t \mid \Phi(s_t)) \, V^\pi(s'_t)$$

and

$$Q_\Phi^\pi(\Phi(s_t), a_t) := \sum_{s'_t \in \{s_t\}^\Phi} P^\pi(s'_t \mid \Phi(s_t)) \, Q^\pi(s'_t, a_t),$$

where $P^\pi(\cdot \mid \Phi(s_t))$ denotes the on-policy distribution over ground states conditioned on the representation value (Definition 4).

These quantities should be understood as theoretical objects that characterize the information retained by the representation under policy $\pi$. They are not assumed to be explicitly computed by the agent.

When the representation is causal (i.e., Markov), these conditional expectations collapse to the original value functions.

**Lemma 1.** *Let $\Phi$ be a Causal (Markov) State Representation (Definition 5), then*

$$Q^\pi(s_t, a_t) = Q^\pi(s'_t, a_t) = Q_\Phi^\pi(\Phi(s_t), a_t) \qquad and \qquad V^\pi(s_t) = V^\pi(s'_t) = V_\Phi^\pi(\Phi(s_t)) \tag{3}$$

*for all $a \in \mathcal{A}$, $s \in \mathcal{S}$, and, $s' \in \{s\}^\Phi$.*

Lemma 1 formalizes the defining property of causal state representations: all states within an equivalence class are behaviorally indistinguishable with respect to rewards and transitions, rendering value functions invariant within the class.

We now define the advantage function induced by a state representation $\Phi$.

**Definition 9** (Advantage under $\Phi$). Given a state representation $\Phi$, the advantage function under $\Phi$ is defined as

$$A_\Phi^\pi(s_t, a_t) := Q^\pi(s_t, a_t) - \bar{V}^\pi(\Phi(s_t)).$$

Note that there is an inherent asymmetry in Definition 9: the state-action value $Q^\pi(s_t, a_t)$ is defined at the level of full states, while the baseline $\bar{V}^\pi(\Phi(s_t))$ aggregates over the equivalence class induced by $\Phi$.

This asymmetry arises because in most policy gradient methods $Q^\pi(s_t, a_t)$ is estimated from Monte Carlo rollouts for each ground state $s_t$ and action $a_t$, making it independent of the representation $\Phi(s_t)$. In contrast, $V_\Phi^\pi(\Phi(s_t))$, modeled using a function approximator, aggregates values over all states in the equivalence class $\{s_t\}^\Phi$, effectively marginalizing out the variables not preserved by $\Phi$. As a result, $A_\Phi^\pi$ quantifies the deviation of a specific state-action pair from the expected value given the agent's state representation.

## 5.2 The scaling effect of the advantage function

We now characterize how the advantage function reweights state-action pairs within a representation-induced equivalence class.

**Theorem 1.** *Let $\Phi$ be an arbitrary state representation. The advantage function can be expressed as*

$$A_\Phi^\pi(s_t, a_t) = \left(1 - P^\pi(s_t, a_t \mid \Phi(s_t))\right)\left(Q^\pi(s_t, a_t) - \tilde{Q}_\Phi^\pi(\neg\langle s_t, a_t\rangle)\right), \tag{4}$$

*where*

$$\tilde{Q}_\Phi^\pi(\neg\langle s_t, a_t\rangle) := \frac{\sum_{s_t', a_t' \neq s_t, a_t} P^\pi(s_t', a_t' \mid \Phi(s_t)) \, Q^\pi(s_t', a_t')}{\sum_{s_t', a_t' \neq s_t, a_t} P^\pi(s_t', a_t' \mid \Phi(s_t))} \tag{5}$$

*is the complementary baseline, and*

$$P^\pi(s_t, a_t \mid \Phi(s_t)) := \pi_\Phi(a_t \mid \Phi(s_t)) \, P^\pi(s_t \mid \Phi(s_t))$$

*denotes the on-policy probability of observing the state-action pair $\langle s_t, a_t\rangle$ conditioned on $\Phi(s_t)$.*

Theorem 1 decomposes the advantage into two components:

1. **Scaling term:** $1 - P^\pi(s_t, a_t \mid \Phi(s_t))$, which downweights the contribution of frequently observed state-action pairs while amplifying rare ones. This reweighting mitigates biases introduced by the policy and reduces the dominance of high-probability pairs, helping to break spurious correlations.

2. **Contrastive term:** $Q^\pi(s_t, a_t) - \tilde{Q}_\Phi^\pi(\neg\langle s_t, a_t\rangle)$ measures how much better or worse a state-action pair is relative to the complementary set in the same equivalence class. The complementary baseline $\tilde{Q}_\Phi^\pi$ is defined as the conditional expectation over all other pairs $(s', a') \neq (s_t, a_t)$. While changing $\Pr^\pi(s_t, a_t \mid \Phi(s_t))$ renormalizes the remaining distribution, the effect on $\tilde{Q}_\Phi^\pi$ is not determined by this change alone, as it depends on how probability mass is redistributed among the other pairs.

**Corollary 1.** *Let $\Phi$ be a Causal (Markov) State Representation. Then the advantage function can be written as*

$$A_\Phi^\pi(s_t, a_t) = \left(1 - \pi_\Phi(a_t \mid \Phi(s_t))\right)\left(Q^\pi(s_t, a_t) - \tilde{Q}_\Phi^\pi(s_t, \neg a_t)\right), \tag{6}$$

*where*

$$\tilde{Q}_\Phi^\pi(s_t, \neg a_t) := \frac{\sum_{a_t' \neq a_t} \pi_\Phi(a_t' \mid \Phi(s_t)) \, Q^\pi(s_t, a_t')}{\sum_{a_t' \neq a_t} \pi_\Phi(a_t' \mid \Phi(s_t))}. \tag{7}$$

This follows directly from Theorem 1 and Lemma 1. In this case, all state-dependent scaling vanishes because the representation $\Phi$ already captures the true causal factors.

## 5.3 Impact on policy gradients

The considerations discussed above would be irrelevant if policy updates strictly followed the exact policy gradient, since the advantage function does not change the expected gradient (Sutton & Barto, 2018). However, as discussed in Section 4.7, in practice policy gradients are estimated using sample-based stochastic gradients.

Consequently, for a given representation $\Phi(s_t)$, it is often the case that all samples in a mini-batch correspond to a single ground state $s_t$, rather than covering all states $s_t' \in \{s\}^\Phi$. When updates rely directly on $Q$-values instead of advantages $A_\Phi^\pi$, this can lead to overly aggressive gradient steps (Chung et al., 2021), causing the agent to reinforce spurious correlations and develop habits that fail to generalize. The effect compounds as the policy reinforces frequently observed state-action pairs, creating a feedback loop (Mei et al., 2022). In the extreme case of a near-deterministic policy, some state-action pairs may be visited so rarely that correcting overestimation errors would require an infeasibly large number of samples.

In contrast, the advantage function $A_\Phi^\pi$ mitigates this problem by scaling gradients according to the probability of each state-action pair (Theorem 1). Gradients for frequently sampled pairs are moderated, while gradients for less common but informative pairs are amplified. This mechanism helps the agent avoid overfitting to spurious correlations and encourages learning based on the causal factors relevant to the task.

### 5.4 Practical considerations

One alternative to using advantages is to increase the batch size, ensuring that each batch more fully covers all states within an equivalence class. However, the batch size required to achieve this may be prohibitively large, depending on the environment. Using advantage functions is generally a more practical and effective solution.

It is also important to note that Suau et al. (2024) report that PPO often struggles with out-of-trajectory generalization, even though default implementations already use advantages. We hypothesize that this limitation arises because, in most PPO implementations, advantages within each mini-batch are normalized before the network update. This normalization removes the scaling effect described in Theorem 1, reducing the ability of the advantage function to counteract policy confounding. Both aspects—batch coverage and advantage normalization—are further analyzed in the experiments section.

### 5.5 Numerical Example: Key2Door

To illustrate the effect of the advantage function in mitigating policy confounding, we revisit the Key2Door environment (Section 2, Figure 1). Suppose the agent's state representation includes only the location variable, $\Phi(s) = l$.

Table 1 presents the $Q$-values and corresponding advantages $A_\Phi^\pi$ when the agent is at location 6,[1] both with $(X = 1)$ and without the key $(X = 0)$, under five different policies. These policies differ in the probability of selecting the optimal action, taking the values 0.5, 0.6, 0.7, 0.8, and 0.9. This allows us to isolate the effect of increasing policy determinism on the distribution of visited state–action pairs. The last two columns indicate the probability that the agent has $P^\pi(X = 1 \mid L = 6)$ or does not have $P^\pi(X = 0 \mid L = 6)$ the key at location 6. [2]

As revealed by Theorem 1, the magnitude of the advantage depends on the joint probability of visiting a specific state-action pair. In particular:

- The advantage of moving right $(A = 1)$ *with* the key decreases as the probability of having the key increases. For example, when $P^\pi(X = 1 \mid L = 6) = 0.832$, the advantage is 0.305, but when $P^\pi(X = 1 \mid L = 6) = 0.999$, it drops to 0.006, even though the corresponding $Q$-value remains 1.

- Conversely, the advantage of moving right $(A = 1)$ *without* the key increases in magnitude as the probability of not having the key decreases. For instance, it is $-0.695$ when $P^\pi(X = 0 \mid L = 6) = 0.168$, but $-0.994$ when $P^\pi(X = 0 \mid L = 6) = 0.000$, while the $Q$-value remains 0.

---

[1]Policy confounding arises at multiple locations in this environment (specifically $L \in \{3, 4, 5, 6\}$) where the agent's location becomes correlated with having collected the key. We focus on $L = 6$ because it is closest to the door, which makes the numerical analysis easiest to interpret and also makes confounding particularly pronounced. In fact, even under a fully random policy, reaching $L = 6$ is already highly correlated with having collected the key.

[2]$Q$-values and advantages are computed using value iteration. $P^\pi(X \mid L = 6)$ is estimated by simulating the policy over multiple episodes.

Table 1: $Q$-values, advantages, and probabilities of key and no key when the agent is at location 6 under five different policies. The agent's state representation consists solely of the location variable.

| | $Q$-value | | | | Advantage $\Phi(s) = l$ | | | | $P^\pi(X \mid L = 6)$ | |
| | No Key | | Key | | No Key | | Key | | No Key | Key |
| | Left | Right | Left | Right | Left | Right | Left | Right | | |
|---|---|---|---|---|---|---|---|---|---|---|
| $\pi(a^* \mid s) = 0.5$ | 0.038 | 0 | 0.662 | 1 | -0.657 | -0.695 | -0.033 | 0.305 | 0.168 | 0.832 |
| $\pi(a^* \mid s) = 0.6$ | 0.274 | 0 | 0.839 | 1 | -0.608 | -0.882 | -0.043 | 0.118 | 0.069 | 0.931 |
| $\pi(a^* \mid s) = 0.7$ | 0.504 | 0 | 0.905 | 1 | -0.456 | -0.960 | -0.055 | 0.040 | 0.018 | 0.982 |
| $\pi(a^* \mid s) = 0.8$ | 0.664 | 0 | 0.934 | 1 | -0.321 | -0.985 | -0.051 | 0.015 | 0.003 | 0.997 |
| $\pi(a^* \mid s) = 0.9$ | 0.759 | 0 | 0.950 | 1 | -0.235 | -0.994 | -0.044 | 0.006 | 0.000 | 0.999 |

The above is important because, as the policy improves, high-probability state-action pairs such as $\langle L = 6, X = 1, A = 1 \rangle$ dominate the training batches, while less frequent but informative pairs, e.g., $\langle L = 6, X = 0, A = 1 \rangle$, are underrepresented. Training on raw $Q$-values treats all observed pairs equally, which can lead the agent to ignore the key variable $X$ and exploit spurious correlations between location 6 and having the key. Notably, even under a random policy (top row of Table 1), the probability of having the key at location 6 is already 0.832, so the agent often receives a reward of 1 simply by moving right, without explicitly reasoning about whether it has the key.

In contrast, training with the advantage function $A^\pi_\Phi$ counteracts this overrepresentation. For example, under the random policy, the advantage of moving right *without* the key is strongly negative ($-0.695$), while moving right *with* the key has a moderate positive advantage (0.305). These differences in magnitude effectively reweight the gradient updates, emphasizing rare but informative state-action pairs and discouraging the agent from developing habits that rely solely on location. As a result, the agent is more likely to attend to the underlying causal factor, i.e., whether it has the key, rather than overfitting to the spurious correlation with location.

## 6 Experiments

The experiments aim to verify whether the insights discussed in the previous section hold in practice. Specifically, we seek to demonstrate that training on the advantage function, rather than raw $Q$-values, helps agents develop state representations that better capture causal factors and thus generalize out-of-trajectory. To test this, we conduct experiments on three gridworld environments: the Key2Door environment described in Section 2, as well as the Frozen T-Maze and Diversion environments introduced by Suau et al. (2024).

We evaluate the agent's performance in both the training environments and in modified versions, referred to as the evaluation environments, where, like in the Key2Door environment, the agent is forced to deviate from its usual trajectory. Furthermore, we analyze the effects of advantage normalization and batch size, which, as discussed in Section 5.4, can influence out-of-trajectory generalization. Finally, we inspect the state representations learned by the agents by measuring the KL divergence of the policies between various state observations. Details about the T-Maze and Diversion environments are provided in Appendix C.

### 6.1 Experimental setup

Agents are trained using two different on-policy policy-gradient methods, REINFORCE (Williams, 1992) and PPO (Schulman et al., 2017) to maximize either the advantage or the $Q$-value. We implement policies and value functions as feedforward neural networks and use a stack of past observations as input in environments that require memory. The results are averaged over 10 random seeds. We report the average return as a function of the number of training steps. The shaded areas show the standard error of the mean. Training is interleaved with periodic evaluations in the original environments and their variants. Further details about the experimental setup are provided in Appendix D.

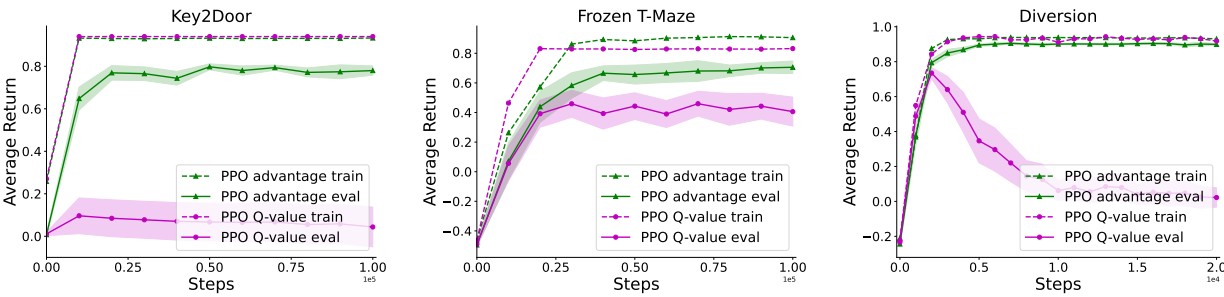

Figure 2: Performance of PPO using Q-values and Advantages in both the training and evaluation variants of the Key2Door (first plot), Frozen T-Maze (second plot), and Diversion (third plot).

## 6.2 Results

Figure 2 shows the performance curves in all three environments for PPO. Agents trained using the advantage function (green) perform well in both the training and evaluation environments, indicating robust out-of-trajectory generalization. In contrast, agents trained on the $Q$-value (magenta) perform poorly in the evaluation environments, suggesting that they overfit to correlations present along the training trajectories and fail when those correlations are broken. Similar results for REINFORCE are reported in Appendix B.1.

The plot on the left of Figure 3 reveals how, as discussed in Section 5.4, normalizing the advantages removes their scaling effect and results in agents being unable to perform well on the evaluation environment. The plot on the right, on the other hand, shows how the performance of policies trained on the $Q$ value improves as we increase the batch size, suggesting that the problem of state-action pair imbalance can sometimes be partly mitigated by using larger batch sizes. Results for the other two environments are provided in Appendices B.2 and B.3.

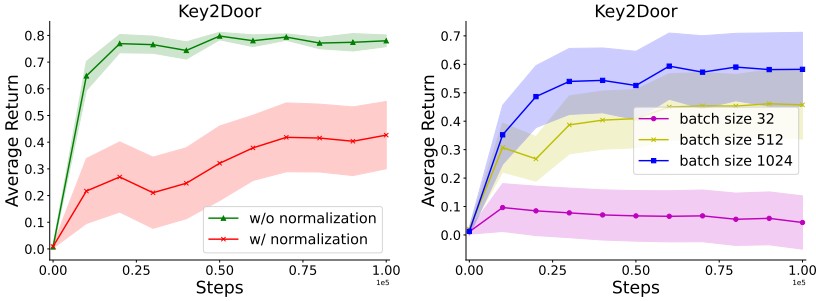

Figure 3: **Left:** Performance of PPO with and without advantage normalization in the Key2Door evaluation environment. **Right:** Performance of PPO with different batch sizes in the Key2Door evaluation environment.

The heatmaps in Figure 4 show the KL divergence between the action distributions when the agent has the key, $\pi(\cdot \mid L = l, X = 1)$, and when the agent does not have the key, $\pi(\cdot \mid L = l, X = 0)$ at each location $l$, measured at different training steps. Because these two states differ only in the key variable $X$, the KL divergence isolates the extent to which the policy's behavior depends on the key rather than on location alone. Higher KL divergence therefore

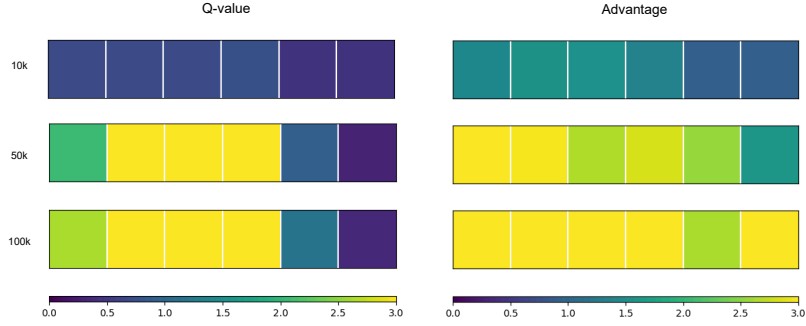

Figure 4: KL divergence of action probabilities with and without the key, measured at different training steps for agents trained on the $Q$-value and the Advantage function.

indicates that the policy conditions its actions on the presence of the key, whereas low KL divergence suggests that the policy largely ignores $X$ and instead relies on spurious correlations induced by the training trajectory.

The heatmap on the left corresponds to agents trained on the $Q$-value, while the one on the right corresponds to agents trained on the advantage function. The results show that policies trained on raw $Q$-values tend to collapse to similar action distributions regardless of whether the agent has the key, indicating policy confounding. In contrast, policies trained with advantages maintain distinct action distributions across key and no-key states, suggesting that the key remains a behaviorally relevant variable throughout training. Similar results for the Frozen T-Maze and Diversion environments are provided in Appendix B.4.

## 7 Limitations

Our analysis assumes discrete state and action spaces, although the results could be extended to continuous domains. In Section 5.1, when defining the advantage function under a given state representation $\Phi$ (Definition 9), we assume that the value function $Q^\pi(s_t, a_t)$ is estimated with respect to the full state, while $V^\pi(\Phi(s))$ is defined for the specific state representation. This assumption is justified in settings where $Q^\pi$ is at least partly estimated from Monte Carlo rollouts, as is common in many actor-critic methods. However, if $Q$ were instead learned entirely via function approximation, then it too would be subject to $\Phi$, and Theorem 1 would no longer hold in its current form.

The experiments were conducted in the same three environments introduced by Suau et al. (2024), which were specifically designed to expose the phenomenon of policy confounding. While these environments are deliberately simple to facilitate analysis, this simplicity limits the generalizability of our findings. As such, we draw no conclusions about the effectiveness of the advantage function for learning causal representations in more complex or high-dimensional domains. Addressing this question would require further empirical investigation and is left for future work. Nevertheless, we believe that the emergence of policy confounding in such minimal settings suggests that the issue may be at least as severe in more complex environments, and that the mechanism identified in this work, namely, the reweighting induced by the advantage function, should transfer to richer domains where similar distributional biases arise.

Finally, throughout the paper, we have taken care not to make strong claims about the effectiveness of the advantage function in learning causal state representations. Rather, we argue that the advantage function can mitigate policy confounding and support the formation of more causally grounded representations. However, its use provides no guarantees that the resulting representations will be truly causal.

## 8 Conclusion

In this paper, we analyzed the role of the advantage function in helping agents learn causal state representations. We showed that the advantage function scales the gradients by the complement of the probability of the corresponding state-action pair. This increases the magnitude of the gradients for state-action pairs that are less likely under the current policy while decreasing it for those that are more likely. As a result, it downweights the impact of state-action pairs that are overrepresented in training batches while amplifying the impact of those that are underrepresented. This helps break spurious correlations introduced by the policy, allowing agents to focus on the true causal factors. Section 5.5 provides a detailed numerical example illustrating this effect.

Our experiments on the Key2Door, Frozen T-Maze, and Diversion environments confirmed that training on advantages leads to more robust agents that generalize better out-of-trajectory. Furthermore, as explained in Section 5.4, our empirical results reveal how implementation choices, such as batch size and advantage normalization, affect the learned representations. Finally, the KL-divergence analysis of the action probabilities further demonstrates that using the advantage function makes agents more reliant on causal factors.

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

# A  Proofs

**Theorem 1.** *Let $\Phi$ be an arbitrary state representation. The advantage function can be expressed as*

$$A_\Phi^\pi(s_t, a_t) = \big(1 - P^\pi(s_t, a_t \mid \Phi(s_t))\big)\Big(Q^\pi(s_t, a_t) - \tilde{Q}_\Phi^\pi(\neg\langle s_t, a_t\rangle)\Big), \tag{4}$$

*where*

$$\tilde{Q}_\Phi^\pi(\neg\langle s_t, a_t\rangle) := \frac{\sum_{s_t', a_t' \neq s_t, a_t} P^\pi(s_t', a_t' \mid \Phi(s_t))\, Q^\pi(s_t', a_t')}{\sum_{s_t', a_t' \neq s_t, a_t} P^\pi(s_t', a_t' \mid \Phi(s_t))} \tag{5}$$

*is the complementary baseline, and*

$$P^\pi(s_t, a_t \mid \Phi(s_t)) := \pi_\Phi(a_t \mid \Phi(s_t))\, P^\pi(s_t \mid \Phi(s_t))$$

*denotes the on-policy probability of observing the state-action pair $\langle s_t, a_t\rangle$ conditioned on $\Phi(s_t)$.*

*Proof.*

$$
\begin{aligned}
A_\Phi^\pi(s_t, a_t) &= Q^\pi(s_t, a_t) - V^\pi(\Phi(s_t)) \\
&= Q^\pi(s_t, a_t) - \sum_{s_t' \in \{s_t\}^\Phi} P^\pi(s_t' \mid \Phi(s_t)) V^\pi(s_t') \\
&= Q^\pi(s_t, a_t) - \sum_{s_t' \in \{s_t\}^\Phi} P^\pi(s_t' \mid \Phi(s_t)) \sum_{a' \in \mathcal{A}} \pi(a' \mid s_t') Q^\pi(s_t', a') \\
&= Q^\pi(s_t, a_t) - \sum_{s_t' \in \{s_t\}^\Phi, a_t' \in \mathcal{A}} P^\pi(s_t', a_t' \mid \Phi(s_t)) Q^\pi(s_t', a_t') \\
&= (1 - P^\pi(s_t, a_t \mid \Phi(s_t))) Q^\pi(s_t, a_t) - \sum_{s_t', a_t' \neq s_t, a_t} P^\pi(s_t', a_t' \mid \Phi(s_t)) Q^\pi(s_t', a_t') \\
&= (1 - P^\pi(s_t, a_t \mid \Phi(s_t))) Q^\pi(s_t, a_t) - (1 - P^\pi(s_t, a_t \mid \Phi(s_t))) \tilde{Q}^\pi(\neg\langle s_t, a_t\rangle) \\
&= (1 - P^\pi(s_t, a_t \mid \Phi(s_t)))(Q^\pi(s_t, a_t) - \tilde{Q}^\pi(\neg\langle s_t, a_t\rangle)).
\end{aligned}
\tag{8}
$$

since

$$
\begin{aligned}
\tilde{Q}^\pi(\neg\langle s_t, a_t\rangle) &= \frac{\sum_{s_t', a_t' \neq s_t, a_t} P^\pi(s_t', a_t' \mid \Phi(s_t)) Q^\pi(s_t', a_t')}{\sum_{s_t', a_t' \neq s_t, a_t} P^\pi(s_t', a_t' \mid \Phi(s_t))} \\
&= \frac{\sum_{s_t', a_t' \neq s_t, a_t} P^\pi(s_t', a_t' \mid \Phi(s_t)) Q^\pi(s_t', a_t')}{1 - P^\pi(s_t, a_t \mid \Phi(s_t))}.
\end{aligned}
\tag{9}
$$

$\square$

# B  Experimental results

## B.1  Results using REINFORCE

Figure 5 shows the performance using REINFORCE in the training and evaluation variants of the three environments.

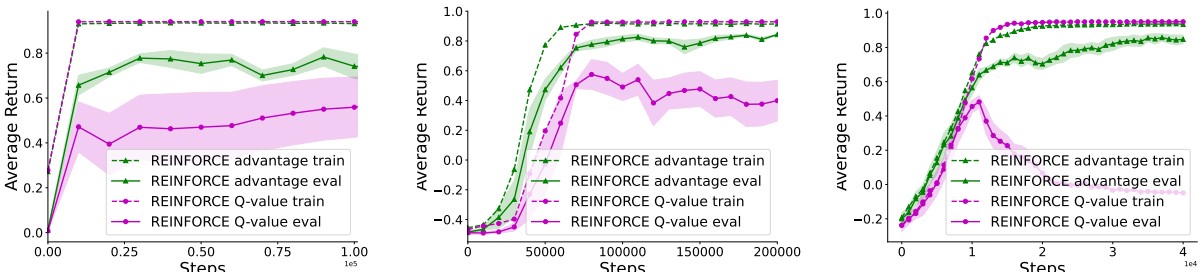

Figure 5: Performance of REINFORCE using Q-values and Advantages in both the training and evaluation variants of the Key2Door (first plot), Frozen T-Maze (second plot), and Diversion (third plot) environments.

## B.2 Advantage normalization

Figure 6 compares the performance of PPO with and without advantage normalization in the evaluation variants of the three environments.

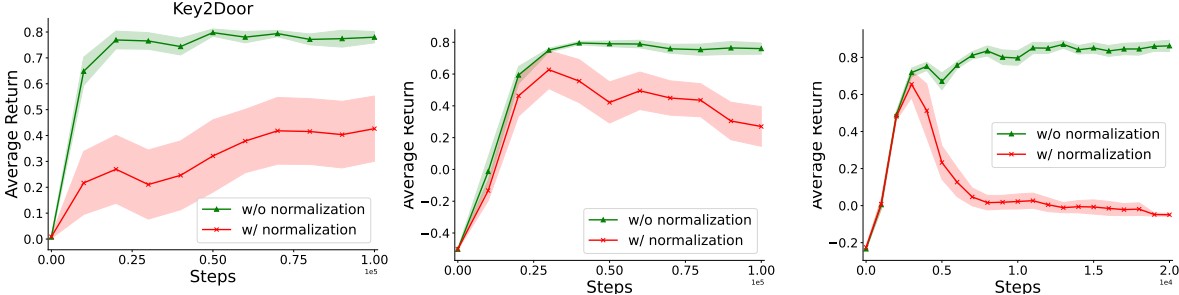

Figure 6: Performance of PPO with and without advantage normalization in the evaluation variants of the Key2Door (first plot), Frozen T-Maze (second plot), and Diversion (third plot) environments.

## B.3 Batch size

Figure 7 compares the performance of PPO with different batch sizes in the evaluation variants of the three environments. While increasing the batch size seems to help in the Key2Door and Diversion environments, it has little effect in the Frozen T-Maze environment.

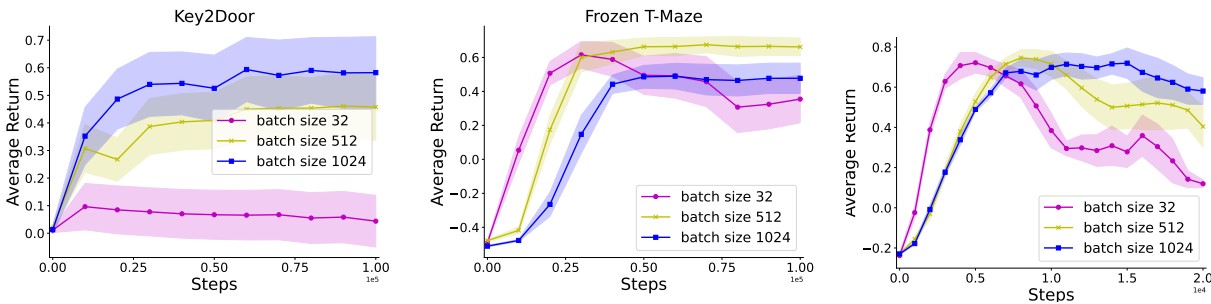

Figure 7: Performance of PPO with different batch sizes in the evaluation variants of the Key2Door (first plot), Frozen T-Maze (second plot), and Diversion (third plot) environments.

## B.4 Policy KL divergence

**Frozen T-Maze** Figure 8 shows the KL divergence of action probabilities in the Frozen T-Maze environment, produced by the policy when the signal is either purple or green, measured at different training steps (top: 10k steps, middle: 50k steps, bottom: 100k steps). To compute these divergences, we take the observation stack received by the agent and query the policy network twice: once with the original stack, and once with the signal bit flipped to the opposite value. The KL divergence at each cell is the average over 100 evaluation episodes.

The left heatmaps reveal that after 100k training steps, the agent trained on the $Q$-value largely ignores the signal, except at the starting location. In contrast, the agent trained on the advantage function conditions its action choices on the signal value throughout the maze.

Interestingly, the left heatmaps also reveal that as training progresses, the trajectories followed by the $Q$-value-trained agent become increasingly deterministic. By the end of training, the agent consistently chooses the top path when the signal is green and the bottom path when the signal is purple. In contrast, the agent trained on the advantage function continues to follow a diverse set of trajectories and does not exhibit a strong preference for any particular path. Note that multiple optimal paths exist for each signal value.

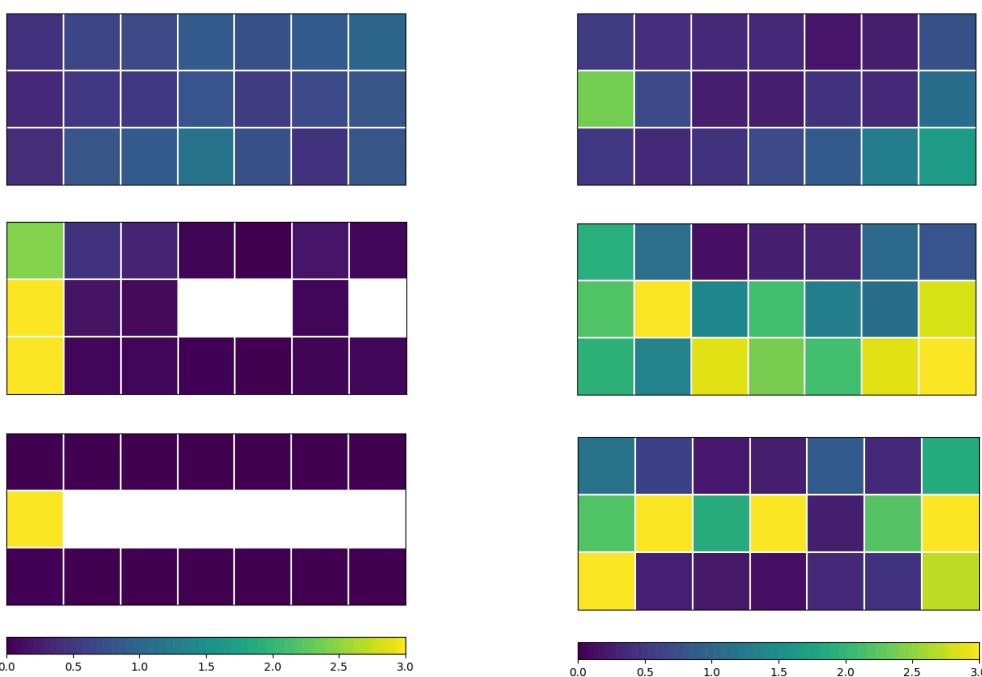

Figure 8: KL divergence of action probabilities in the Frozen T-Maze environment at different training steps (top 10k steps, middle 50k steps, and bottom 100k steps) for agents trained on the $Q$-value (left) and the Advantage function (right).

**Diversion** Figure 9 shows the policy KL divergence across each column in the Diversion environment, measured at different training steps (top: 3k steps, middle: 10k steps, bottom: 20k steps). For each column, the divergence is computed by comparing the action probabilities output by the agent's policy when positioned in the top versus bottom row.

The left heatmaps indicate that the agent trained on the $Q$-value learns to ignore the bit indicating the row after 10k training steps. In contrast, the agent trained on the advantage function continues to use the row information when deciding which action to take.

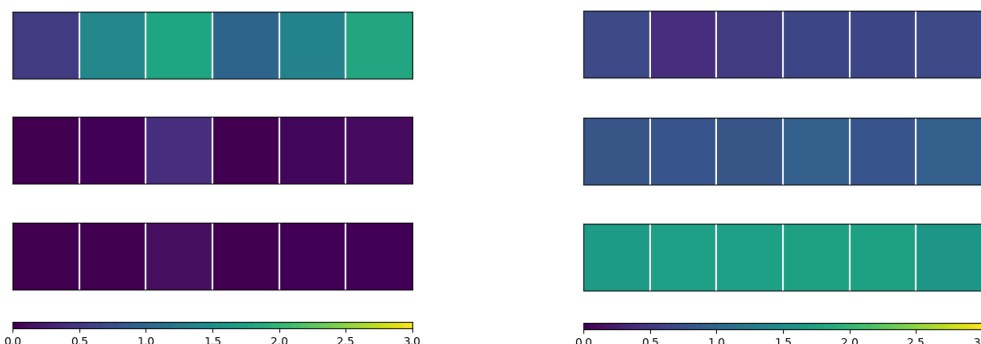

Figure 9: KL divergence of action probabilities in the Diversion environment measured at different training steps (top 3k steps, middle 10k steps, and bottom 20k steps) for agents trained on the $Q$-value (left) and the Advantage function (right).

## C  Environments

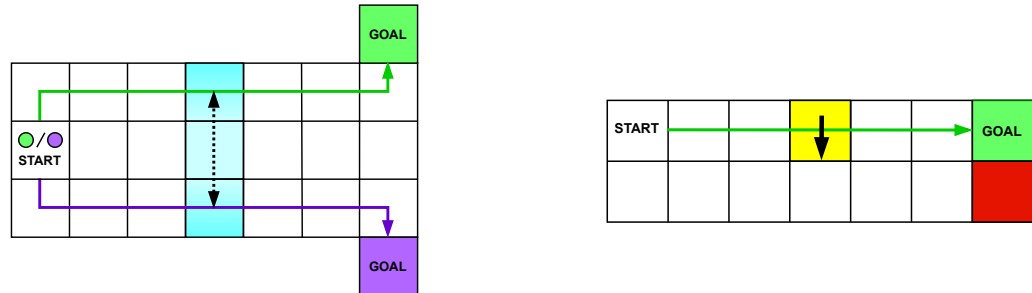

Figure 10: Illustrations of the Frozen T-Maze (left) and Diversion (right) environments.

**Frozen T-Maze**  This environment is a variant of the popular T-Maze setting (Bakker, 2001). At the starting location, the agent receives a binary signal: either green or purple. Its task is to navigate to the right and reach the correct goal at the end of the maze. The agent receives a reward of $+1$ for reaching the green (purple) goal when the green (purple) signal was observed, and a penalty of $-1$ otherwise. Additionally, a time penalty of $-0.01$ is applied at each timestep to encourage the agent to take the most direct path to the goal.

At every timestep, the agent observes its current location within the maze, represented by a one-hot-encoded vector. However, the initial signal, represented by a binary variable, is only provided to the agent at the starting location. Crucially, the agent is capable of remembering past observations. When moving randomly, it must retain the initial signal throughout its trajectory. However, once it learns the shortest path to each goal (illustrated by the green and purple arrows), the agent can safely disregard the initial signal. This is because the agent can infer the signal based on its location: if it is on the green (purple) path, it must have received the green (purple) signal. Note that the two paths highlighted in Figure 10 are not the only optimal ones. However, for the agent to disregard the initial signal, the paths must not overlap.

We train agents in the original environment and evaluate them in a modified version where an icy surface (shown in blue) is introduced in the middle of the maze. This ice causes the agent to slip between the upper and lower cells.

**Diversion**  In this environment, the agent must move from the start state to the goal state shown in Figure 10 (right). A reward of $+1$ is given for reaching the goal, and a penalty of $-1$ is incurred if the agent reaches the red cell instead. As in the other environments, there is a per-timestep penalty of $-0.01$. Observations are 8-dimensional binary vectors: the first 7 elements indicate the column where the agent is located, and the last element indicates the row.

After the agent learns the optimal policy (shown by the green arrow), it can ignore the last element of the observation vector. This is because the optimal policy never visits the bottom row. We train the agent in the original environment and evaluate it in a modified version containing a yellow diversion sign in the middle of the maze, which forces the agent to move to the bottom row.

## D   Experimental setup

The experiments were run on a laptop equipped with an Apple M2 Pro processor (12 cores) and 16 GB of RAM. Each run took less than 5 minutes and used at most 2% of the total RAM.

Agents were trained using Stable Baselines3 (Raffin et al., 2021). The hyperparameters are listed in Tables 2 (PPO) and 3 (REINFORCE). For PPO, we adopted the hyperparameters used by Schulman et al. (2017) in their Atari experiments, except for the learning rate, which we increased to 1.0e-3 to accelerate convergence. We implemented a minimal version of the REINFORCE algorithm with only three hyperparameters (learning rate, discount factor, and entropy coefficient), for which we used the same values as in PPO. For the Frozen T-Maze, we used a stack of the past 30 observations as input, since solving the task requires memory.

Table 2: PPO hyperparameters.

| | |
|---|---|
| Rollout steps | 128 |
| Batch size | 32 |
| Learning rate | 1.0e-3 |
| Number epoch | 3 |
| Discount $\gamma$ | 0.99 |
| GAE $\lambda$ | 0.95 |
| Entropy coefficient | 1.0e-2 |
| Clip range | 0.1 |
| Value coefficient | 1 |
| Number Neurons 1st layer | 128 |
| Number Neurons 2nd layer | 128 |

Table 3: REINFORCE hyperparameters.

| | |
|---|---|
| Learning rate | 1.0e-3 |
| Discount $\gamma$ | 0.99 |
| Entropy coefficient | 1.0e-2 |
| Number Neurons 1st layer | 128 |
| Number Neurons 2nd layer | 128 |

