# OpenReview forum: "Breaking Habits: On the Role of the Advantage Function in Learning Causal State Representations"
_TMLR — Rejected by TMLR_

### Review · Reviewer_os8x · 2025-12-04

**Summary Of Contributions:**

This work claims that the use of advantage function for RL problems especially on-policy settings helps avoid exploiting spurious correlation between observation and reward, leading to better catch of causal effect, with some math and experimental results.

**Audience:**

Yes

**Audience Explanation:**

The perspective and findings may be potentially very interesting; the way it presents lacks rigor making it difficult for me to judge the credibility of the claim.

**Claims And Evidence:**

No

**Claims Explanation:**

The overall story is simple and clear and should be interesting for the community.
However, I would say that the current presentation of math lacks rigor and the claim is not credible in the current manuscript.

(1) Definition 2 lacks rigor and notations are all unusual (or strange) and mathematically do not make sense.  < > notation is strange too...

(2) the definition of do operator is also not defined rigorously.  Also define this first before definition 5.  And Q, R, Q^\pi,,, sometime are defined for S and sometimes for \bar{S}; again \bar is usually for closure.  Also don't use the same function Q, R etc. if the domains are different (at least do restriction | )

(3) What is Phi^\pi?  Different from Phi?  There are many such notations that are not clearly defined.

(4) What is the definition of P^pi(s_t,a_t | Phi(s_t))?  Only P^pi(s|Phi(s)) is defined.  Q sometimes has Phi subscript but sometimes not.

(5) The authors say ...since it does not depend on P^pi...; no it depends on it right?  If it is large, then other probabilities become smaller.

(6) No theory for the effect of normalization?

(7) Table 1 is very hard to parse.  Where is -0.01 etc etc.

(8) Crucially, is it for causality?  Isn't it just about weighting??  Even with this weighting, it is just random (with better distribution) isn't it?

(9) As the authors write in limitation; if they cannot say that the resulting representations are truly causal, what is the point?  Make sure the work has the clear claims that are well-justified.  Otherwise, present this in the way it is about conjecture.





As for minor issues.

(1) In section 2, the sentences "During training,...This implies that once..." are hard to parse.

(2) What is l_t in page 3?  Difference from L?

(3) "the above conditions" in page 4; what is this referring to?

(4) The paragraph after corollary 1 should be detailed.

(5) It also breaks good habit right?

(6) don't use "the above is key" or "taking the right action" because they are confusing (is it the physical key in the problem?  is it left right action or correct action?)

(7) page 9 typo (represantion); also Figure 4 is hard to parse

**Requested Changes:**

Please address my concerns given above.

---

> ### Author Response · Authors · 2025-12-31
>
> We would like to thank you for your thorough and thoughtful review of our work.
>
> We apologize for the confusion caused by undefined functions and notation. Most of the notation is borrowed from Suau et al. (2024), and some symbols were not properly defined here. We have now explicitly defined all notation and resolved points (1)–(4) and the minor issues listed in your review.  Please let us know if anything remains unclear.
>
> Please find our responses to your questions below:
>
> > (5) The authors say ...since it does not depend on P^pi...; no it depends on it right? If it is large, then other probabilities become smaller.
>
> $\tilde{Q}^\pi$ does not depend on $\Pr^\pi(s_t, a_t \mid \Phi(s_t))$. By construction, it is computed by normalizing over the probabilities of all state–action pairs in the equivalence class excluding $\langle s_t, a_t \rangle$. Consequently, $\tilde{Q}^\pi$ depends only on the relative probabilities among the remaining pairs.
>
> > (6) No theory for the effect of normalization?
>
> We are not entirely certain which theoretical result the reviewer is referring to. Our empirical observation is that standard PPO can still suffer from policy confounding, even though it uses advantages. We note that in most PPO implementations, advantages are normalized within mini-batches before the gradient update. This normalization removes the scaling effect of the advantage function identified in Theorem 1, thereby weakening its ability to counteract policy confounding. We have clarified this discussion in the revision.
>
> > (7) Table 1 is very hard to parse.
>
> We have revised the wording of the entire section to improve clarity and readability. Please let us know if any aspects of the table remain unclear.
>
> > (8) Crucially, is it for causality? Isn't it just about weighting?? Even with this weighting, it is just random (with better distribution) isn't it?
>
> Our claim is not that the advantage function induces causal reasoning. Rather, it provides a principled reweighting mechanism that counteracts policy confounding by downweighting overrepresented state–action pairs (e.g., location 6 with the key) and upweighting rarer but informative pairs (e.g., location 6 without the key). This reweighting corrects for policy-induced distributional shifts and discourages the agent from reinforcing confounded state representations.
>
> > (9) As the authors write in limitation; if they cannot say that the resulting representations are truly causal, what is the point? Make sure the work has the clear claims that are well-justified. Otherwise, present this in the way it is about conjecture.
>
> We do not claim to guarantee fully causal state representations. Instead, we identify policy confounding as a key obstacle to learning policy-invariant (and thus causally meaningful) representations, and we show that advantage-based updates mitigate this obstacle. Our contribution is therefore directional rather than absolute: the advantage function biases learning away from policy-specific correlations and toward representations that are more robust, without claiming full causal correctness.

---

> ### Comment · Reviewer_os8x · 2026-01-02
> **Thank you for the response**
>
> Thank you for the response;
> while I check the details, I would say that the current claim that connects the work to learning causal relation is still a bit misleading.
> Please do not generalize too much the claim you verify.  I understand what the authors are trying to say, but it seems it is more reasonable to say that the work is for how to deal with out-of-distribution issue (title etc. are a bit misleading).

---

> > ### Author Response · Authors · 2026-01-03
> >
> > We agree that claims about learning causal relations must be made carefully, and we have revised the paper to make the scope of our claims more explicit. Our contribution is to show how the advantage function mitigates policy confounding and thereby supports the learning of causal state representations, without claiming full causal discovery. The title deliberately emphasizes *the role* of the advantage function, rather than suggesting that it guarantees or directly recovers causal state representations.
> >
> > Moreover, learning representations that generalize beyond the trajectories induced by a specific policy (out-of-trajectory generalization) is equivalent to capturing the underlying causal factors governing rewards and transitions (Suau et al., 2024, Theorem 1). In this sense, framing the title in terms of causal state representations or out-of-trajectory generalization refers to the same underlying objective.

---

> > > ### Comment · Reviewer_os8x · 2026-01-11
> > > **Thank you for the response**
> > >
> > > The manuscript seems getting improved; but I still have major concerns.  I am sorry if you feel I am not getting it right though.
> > >
> > > (1) The concern (5) above is still a concern.  I may understand what you want to say, but if you say the value is independent of P(s_t, a_t|...), it should mislead readers in several ways; why can you say that changing the value P(s_t, a_t|...) does not break the proportions of other events?  Are you assuming probabilistic independence somewhere?
> > >
> > > (2) \bar{S} is not formally defined; also the updated manuscript somehow "defines" do operator, but what is needed here is more mathematical definition; otherwise what was the point of defining MDP etc. mathematically?
> > >
> > > (3) There still remain several notations that are not defined rigorously before they are introduced.
> > >
> > > (4) I understand that it "may" help capturing causal effect; but (a) if you still use compressed state representation, then what can be best hoped for is a better distribution that actually may produce improved performance when the initial location etc. is randomized for example.  As you say, full representation is necessary in most cases right?  So, the argument that may make sense is, better distribution helps learn NN so that it distinguishes states that are important for causality.  Anyway, causality and what the final goal is is not clearly stated (there are several texts that tells the story, but formal statement is not clear)
> > >
> > > (5) The above "(6) No theory for the effect of normalization?" is for asking for some mathematical explanations for why PPO's normalization may mitigate bad correlation effect.
> > >
> > > Finally, I think the discovery and the story should be valuable, so I strongly recommend summarizing it in more formal way so that readers don't get misguided or over generalize.
> > > If you feel my review is missing some points, I apologize in advance, but I would recommend resubmitting after some modification.

---

> > > > ### Author Response · Authors · 2026-01-11
> > > >
> > > > Thank you for the follow-up. We address each point below.
> > > >
> > > > (1) We agree with the concern and have clarified in the current revision that, while changes in $\Pr^\pi(s_t, a_t \mid \Phi(s_t))$ induce a renormalization of the remaining conditional distribution, the resulting change in the complementary baseline $\tilde{Q}$ is not identifiable from this marginal change alone. In other words, $\tilde{Q}$ depends on the redistribution of probability mass among the other state-action pairs in the equivalence class, and knowing only the change in $\Pr^\pi(s_t, a_t \mid \Phi(s_t))$ does not predict how $\tilde{Q}_\Phi^\pi$ changes. This clarification is now explicitly stated in the text.
> > > >
> > > > (2) $\bar{S}$ is the projected state space defined in Definition 3 as part of the state representation. The do-operator is a standard concept in causal inference (Pearl, 2016). Providing a formal definition would likely overwhelm the text and distract from the main points, so we assume familiarity with this common notation.
> > > >
> > > > (3) We would appreciate it if the reviewer could point out the specific notations that appear insufficiently defined.
> > > >
> > > > (4)
> > > >
> > > > > what can be best hoped for is a better distribution that actually may produce improved performance
> > > >
> > > > A "better" (more uniform) distribution over state-action pairs is key to improving generalization and supporting causal reasoning. This aligns with standard causal inference techniques, such as inverse probability weighting, which adjust observational data to approximate a randomized controlled trial.
> > > >
> > > > > The argument that may make sense is, better distribution helps learn NN so that it distinguishes states that are important for causality. Anyway, causality and what the final goal is is not clearly stated.
> > > >
> > > > We respectfully disagree. The argument the reviewer mentions is already explicitly stated in the paper. In particular, it appears in the Introduction (Contributions paragraph), at the beginning of Section 5, and in the Conclusion, where we describe how the advantage function reweights state-action pairs to mitigate policy confounding and allow the agent to focus on causally relevant factors.
> > > >
> > > > (5) Our argument is in fact the opposite: mini-batch normalization removes the probability-dependent scaling effect of the advantage function, which is precisely the mechanism that counteracts policy-induced imbalance. As a result, PPO with advantage and normalization fails to mitigate policy confounding, leading to the poor performance observed in evaluation environments (see Section 5.4 and Figure 3, left).
> > > >
> > > > **We respectfully note that, while the reviewer raises some valid points, the concerns primarily relate to clarity and presentation. None of these affect the correctness of our theoretical results or the empirical claims of the paper.**

---

### Review · Reviewer_j4MH · 2025-12-10

**Summary Of Contributions:**

This paper proposes using the advantage function instead of Q-values directly in reinforcement learning, such that the learned behavior is more causal and, thus, more accurate.

In particular, the paper stresses how this helps with overcoming policy confounding, a particular type of confounding in reinforcement learning where policies favor actions in a state not only because of the true causal mechanisms, but also because these states are only reached in particular ways following the policy itself. (I.e., important features might be ignored as they are confounded by the trajectories generated by the policy.)

Experiments on simple environments are included, demonstrating how the advantage function helps overcome policy confounding.

**Additional Comments:**

Questions

- Can you please elaborate on why using larger batch sizes helps with policy confounding as well? Wouldn't less likely values always remain underrepresented? Is it more about these unlikely states appearing in a batch at all?
- How is "slipping" implemented in the "Frozen T-Maze" environment?
- In Figure 7, why does batch size 32 peak so high for Diversion? And why is batch size 512 optimal in Frozen T-Maze?

Minor Points

- From how I understand the Key2Door experiments, the game is always terminated once the door is reached, even if there is no key, correct? In any case, this should be clarified, as it can cause confusion, for example, in Table 1, where the Q-Values for "No Key", "Right" would not be 0 if the environment didn't terminate.
- In Section 2, the states are called $L \in \{0,1,\dots,6\}$ but Figure 1 has states 1 to 6 (presumably 7 if you would view door as the next area after 6).
- In Section 6, "section 5.4" is written without capitalizing the "s".
- Adding the game title above the plots in Figure 2, and maybe also other figures, would improve clarity a bit.
- Figure 2 removes a bit of the "Average Return" text for the second and third plot (it is clipped by a bit).
- It would be nice to somehow include an advantage function reference in Figure 3 (right plot). Maybe using a small batch size. Do you think that makes sense?
- In Figure 4, adding the steps to the figure itself, not just the caption, would be better for readability. Same for being trained on the Q-value or advantage function. The text of the legend could also be increased a bit.
- Figure 4 flows onto page 10.

**Audience:**

Yes

**Audience Explanation:**

The presentation of the paper is very good, and the role the advantage function can play in overcoming policy confounding is relevant for the reinforcement learning community.

From a purely causal point of view, overcoming policy confounding certainly makes the method "more causal". There are, of course, no guarantees that any learned behavior will be fully causal, and it might still suffer from other types of confounding (also quoting the paper: "its use provides no guarantees that the resulting representations will be truly causal"). As policy confounding seems to be mostly relevant to the reinforcement learning setting, I think that this work will be more interesting to this community than to the rest of the causality community.

**Broader Impact Concerns:**

No Concerns

**Claims And Evidence:**

Yes

**Claims Explanation:**

The three environments (Key2Door, Frozen T-Maze, and Diversion) show the behavior clearly. Using the advantages function reaches better results than standard Q-learning, and this is true across all these environments. The setup is explained well, and the results are clearly understandable in the main paper and appendix.

As the authors recognize, all these environments are rather simple and small, and, quoting the paper, "this simplicity limits the generalizability of our findings". However, I find them to be sufficient for the scope of this paper, which is more about the theoretical contribution on the benefit of using the advantage function to overcome policy confounding.

**Requested Changes:**

In my opinion, these experiments are sufficient for creating an interest in the reinforcement learning community. Naturally, the impact would be increased if this behavior could be recreated in more complex, larger environments. Even without adding new experiments, I would be interested in the authors' thoughts on how more realistic environments might or might not behave similarly with respect to policy confounding and the use of the advantage function. Therefore, new experiments are, to me, not critical, but would strengthen the work.

Overall, I have no critical concerns.

---

> ### Author Response · Authors · 2025-12-31
>
> We thank the reviewer for the thoughtful and constructive feedback, and for recognizing both the clarity of the presentation and the relevance of policy confounding to reinforcement learning.
>
> All issues listed under minor points have been addressed in the revision. Please find our responses to your questions below.
>
> > Can you please elaborate on why using larger batch sizes helps with policy confounding as well? Wouldn't less likely values always remain underrepresented? Is it more about these unlikely states appearing in a batch at all?
>
> We agree with the reviewer’s intuition. Larger batches do not eliminate underrepresentation of unlikely state–action pairs, but they increase the probability that such pairs appear at least occasionally within a batch. This partially mitigates policy confounding by reducing extreme sample imbalance within updates, which aligns with our analysis in Section 5.5 and the batch-size experiments.
>
> > How is "slipping" implemented in the "Frozen T-Maze" environment?
>
> Slipping forces the agent to move two cells in the vertical direction when stepping on ice: downward when the agent is at the top cell and upward when it is at the bottom cell. No slipping occurs when the agent is on the middle row. This ensures that the resulting trajectories remain feasible within the original environment.
>
> > In Figure 7, why does batch size 32 peak so high for Diversion? And why is batch size 512 optimal in Frozen T-Maze?
>
> The peak performance for batch size 32 in the Diversion environment appears comparable to that of batch sizes 512 and 1024, although it occurs earlier during training. Smaller batch sizes result in more frequent parameter updates, which seems to accelerate learning in both the Frozen T-Maze and Diversion environments.
>
> For the Frozen T-Maze, it is difficult to explain why batch size 512 outperforms 1024. Both batch sizes outperform 32, which is consistent with our intuition that larger batch sizes help mitigate policy confounding. Given that the results are averaged over only 10 runs and exhibit high variance, the observed difference between batch sizes 512 and 1024 might be due to stochasticity.
>
> > I would be interested in the authors' thoughts on how more realistic environments might or might not behave similarly with respect to policy confounding and the use of the advantage function. Therefore, new experiments are, to me, not critical, but would strengthen the work.
>
> We agree that extending the analysis to larger or higher-dimensional domains would strengthen the impact. However, the simplicity of the environments is intentional: our goal is to isolate policy confounding and its interaction with the advantage function in controlled settings that allow clear interpretation. Introducing more complex environments would add confounding factors and might distract from the core focus of our study.
>
> The fact that policy confounding arises in such minimal environments suggests that it may be even more pronounced in higher-dimensional domains, where policies can exploit a larger number of spurious correlations.
>
> Importantly, our theoretical results are not restricted to small or discrete state spaces. The scaling effect of the advantage depends on the on-policy distribution induced by the policy, rather than on the size of the state space itself. Consequently, the advantage function should also help mitigate distributional bias introduced by the policy in more complex settings.

---

> > ### Comment · Reviewer_j4MH · 2026-01-12
> >
> > I apologize for the delay in my response and thank the authors for their detailed rebuttal.
> > I have no major concerns or further questions remaining.

---

### Review · Reviewer_Cxxp · 2025-12-15

**Summary Of Contributions:**

The paper studies the issue of policy confounding in RL. In particular, some policies might mis-corelate the state and the reward, leading to unexpected habits. The paper suggests that the advantage function in many RL learning methods not only reduces variance in the learning process, but also contributes to "decorrelation" of such spurious correlations.

As far as I understand, the idea of this work is motivated by two observations: 1) the phenomenon of policy confounding, which exists more extensively in on-policy methods, explored by Suau et al. (2024), and 2) the choice of baselines matters in policy gradient methods, explored by Chung et al. (2021) and Mei et al. (2022).

The new things being developed are mainly sitting in Section 5.

**Strengths**

The problem is well motivated and is indeed important in RL studies. The paper uses a simple example to help illustrate its idea throughout the paper. The storyline with high-level intuition is nicely put.

**Weaknesses**

- Credit: The entire section 4 should cite Suau et al. (2024) more explicitly. It is now written as if Suau et al. (2024) only provide certain observations. However, they have established what you have in section 4.

- Weak justification of arguments. See the next section.

**Additional Comments:**

- I have a more general question: do you also see such a strong policy confounding issue if you have a random starting state in training? I do not think this will remove entirely potential mislearned correlation, but it seems to me that this can help at least the naive implication of “moving to location 3 implies having a key”.

**Audience:**

Yes

**Audience Explanation:**

Quite a few TMLR readers can be interested in RL. This work studies an important issue in RL research. They could benefit from this work once it is further polished.

**Claims And Evidence:**

No

**Claims Explanation:**

### Section 5.5 and Table 1

It’s not clear what you want to convey here. It is, first of all, unclear what $\pi(a^*|s)$ means in this case; why do you care L=6 the most; and why do you want to investigate different probabilities of the optimal action. Also, sometimes you use X=0,1, sometimes you directly use No Key / Key; same for actions: sometimes left /right, sometimes A=0,1. These seemly small things altogether introduce unnecessary barriers.

### Figure 2, PPO with advantage / Q-value

This experiment does not seem directly relevant to the message of the paper. What you claimed was a relation between the advantage function and policy confounding. However, figure 2 itself can indicate something else from this hypothesis.

### KL Divergence

The heat map in Figure 4 and the KL divergence do measure something a bit more interesting than Figure 2 for this paper. However, it will be better to expand on why “Higher KL divergence indicates that the policy places more importance on the key feature.”

### Weak justification of arguments

- Quite some arguments are not strong enough, including the above-mentioned ones.

- Regarding “Strictly speaking, the only representation that satisfies the above conditions is the Markov state representation.” Claiming something to be “the only” typically requires proper justifications, which are currently missing.

- Weak notations also hinder the soundness of the arguments. See below.

### Notations

Without careful notations and arguments, the clarity of the messages you want to convey could be highly compromised. I will list some examples later, but I recommend a more thorough examination throughout the paper.

- The notation of transition switches from $T$ to $P$ at some point. Please make it consistent. Or I assume that by $P$, you mean the probability. However, still, you need to make the notation clear.

- Please check the consistency of the notations $F$ and $\mathcal F$. Also, $F=F^1,F^2,\cdots,F^F$ is a pretty terrible notation. The definition of $F$ is unclear. Make it clear whether it's a tuple, a vector, a set, a subspace, or a value.

- A few notations in Definition 5 are problematic. E.g., You did not seem to have defined $\Phi^\pi$, $R^\pi$, and $S^\pi$. Furthermore, the meanings of them are unclear. None of them, the representation functions, the reward functions, and the state space, seems to have a natural association with a policy. Also, it’s not clear what “$P^\pi$ is a probability under $\pi$” means. It should, for example, be “the probability of … to … under $\pi$”. Otherwise, it’s not clear the spaces between the mapping or the transitions. Therefore, it hinders the clarity of your argument and what follows.

- I think it’s dangerous to use $P^\pi$ as the notation in definition 6. In particular, since $P$ in some parts of your paper, and in RL in general, usually means “transition from the current time step to the next when taking an action”. However, here, I think what you simply want is some sense of “integral” for states in the representation states, which involves only the “mapping” between spaces rather than any “transition”. Hence, it risks the mix and ambiguous meaning in your notations.

- The negation notations, both negating $a_t$ and negating $\langle s,a\rangle$, are not properly defined and illustrated. Also, the meaning of $\langle s,a\rangle$ itself is unclear.

- It’s not clear what it means by “In our case, do(Φ(st)) means setting the variables forming the state representation Φ(st) to a particular value and considering all possible states in the equivalence class, s′ t ∈ {st}Φ,”. In particular, the former part reads rather fine, but it is not clear what you want to “consider”.

**Requested Changes:**

See the boxes before.

---

> ### Author Response · Authors · 2025-12-31
>
> We would like to thank you for your thorough and thoughtful review of our work. Please find our responses to your comments below.
>
> **Credit**
> > The entire section 4 should cite Suau et al. (2024) more explicitly. It is now written as if Suau et al. (2024) only provide certain observations.
>
> Section 4 is the preliminaries section, which is typically reserved for background material on which the paper builds. Nevertheless, we agree that the connection to Suau et al. (2024) should be made more explicit. We have therefore added a paragraph at the beginning of the section clarifying that Section 4 summarizes the necessary background, and that a more detailed discussion can be found in Suau et al. (2024).
>
> **Section 5.5 and Table 1**
>
> > It’s not clear what you want to convey here. It is, first of all, unclear what means in this case and
>
> The purpose of Section 5.5 and Table 1 is to provide a concrete numerical illustration of how the advantage function rescales gradient contributions under a confounded state representation. We have rewritten parts of Section 5.5 and added clarifications to make this interpretation clearer.
>
> > why do you care L=6 the most.
>
> Policy confounding arises in multiple locations in the Key2Door environment (specifically $L \in \{3,4,5,6\} $), where the agent’s location becomes correlated with having collected the key under the training policy, even though the key remains causally relevant for the reward. We focus on $L = 6$ because it is closest to the door, which makes the analysis easier to interpret numerically and also makes confounding particularly pronounced: even under a fully random policy, reaching  $L=6$ is already highly correlated with having collected the key.
>
> > why do you want to investigate different probabilities of the optimal action.
>
> We vary the probability of selecting the optimal action to illustrate how increasing policy determinism exacerbates confounding: as the policy improves, the trajectory distribution concentrates on a small subset of state–action pairs, amplifying biases when learning from raw Q-values and highlighting the corrective role of the advantage.
>
> **Figure 2, PPO with advantage / Q-value**
>
> > This experiment does not seem directly relevant to the message of the paper. What you claimed was a relation between the advantage function and policy confounding. However, figure 2 itself can indicate something else from this hypothesis.
>
> Figure 2 is intended to empirically illustrate the mechanism analyzed in the paper, namely, how advantage-based updates mitigate policy confounding through gradient reweighting.
>
> By comparing advantage-based and $Q$-value–based training within the same on-policy algorithm (PPO), the figure shows that $Q$-based agents fail precisely in evaluation settings designed to break policy-induced correlations (e.g., having the key at location 6), while advantage-based agents remain robust. This behavior is consistent with our theoretical result that advantages downweight frequently occurring state–action pairs.
>
> We agree that performance curves alone can be ambiguous; for this reason, Figure 2 should be read alongside the KL-divergence analysis, which directly probes the predicted scaling effect and its impact on representation usage. Finally, as discussed in the limitations section, we would like to emphasize that we do not claim causality guarantees, only that advantage-based updates reduce policy confounding under the conditions studied.
>
> **KL divergence**
> > The heat map in Figure 4 and the KL divergence do measure something a bit more interesting than Figure 2 for this paper. However, it will be better to expand on why “Higher KL divergence indicates that the policy places more importance on the key feature.
>
> The KL divergence measures how differently the policy behaves when the key variable differs while all other observed variables are held fixed. A high KL divergence implies that the policy is explicitly conditioning its behavior on the key feature, rather than relying solely on location-induced correlations. Conversely, low KL divergence indicates that the policy treats these states similarly and thus largely ignores the key. We have clarified this in the revision.
>
> **Argument justification**
>
> > Regarding “Strictly speaking, the only representation that satisfies the above conditions is the Markov state representation.” Claiming something to be “the only” typically requires proper justifications, which are currently missing.
>
> This result is shown in Suau et al. (2024). We have added the appropriate citation immediately following the statement.
>
> **Notations**
>
> Thank you for carefully reviewing the mathematical notation. We apologize for the confusion. Most of the notation is borrowed from Suau et al. (2024), and some symbols were not properly defined here. We have now defined all notation explicitly and resolved the inconsistencies listed in your review. Please let us know if anything remains unclear.

---

> > ### Comment · Reviewer_Cxxp · 2026-01-13
> >
> > Thank you for the response. However, several questions and issues remain unresolved for me.
> >
> > 1. **Rigor**: First, I share Reviewer os8x's concern regarding rigor. While some notations have been fixed, others remain concerning.
> >     - Overloading Notation: Overloading $F$ and $\mathcal{F}$ for several different meanings creates confusion. For example, writing $|F|$ implies $F$ is a set, yet it is, at the same time, treated as a random variable with corresponding domains.
> >     - Definitions: The definitions of $P^\pi$, $Pr^\pi$, and $T^\pi$ are still not fixed.
> >
> > 2. **Tables and Figures**: Second, I share the concern raised by other reviewers that Table 1 and some figures are difficult to parse. I feel this issue remains in the current revision. While I acknowledge that some information is technically present in the draft, it feels scattered and isolated. I use Table 1 as an example to explain. Just to be clear, I agree with and understand your high-level goal: using the Key2Door example to illustrate the effect of the advantage function in mitigating policy confounding. However, the current text does not clearly show how the table supports that claim.
> >     - The mixed use of notations (e.g., switching between "$X=0$" and "No Key"; "$A=1$" and "Move Right") creates immediate obstacles, but the main difficulty lies in the organization of the argument.
> >     - Just by looking at the table, it is difficult to know where to start or what the numbers actually indicate. For such a dense table, it requires a more detailed walkthrough of a logical progression: first defining the rows and columns, then explaining how to parse the values, and finally explicitly demonstrating how this data serves as evidence for your hypothesis.
> >     - In the text, you state: "Table 1 presents the Q-values and corresponding advantages $A^\pi_\Phi$." I suggest using complete and precise notations here. Since the paper discusses various Q-functions and Advantage functions, using specific notations would make it immediately clear exactly which values you are reporting.
> >     - You mention: "These policies differ in the probability of selecting the optimal action... taking values 0.5... to 0.9...., increasing policy determinism ..." First, the notation $\pi(a^\*|s)$ is not well-specified. It would be clearer to formally define the optimal action $a^\*(s)$, which depends on both the key status $X$ and the location $L$. Otherwise, it's not immediately clear what $a^\*$ is and what $s$ is. Also, you can move the following info earlier: "$\pi(a^\*|s) = 0.9$ represents a 'Good Policy' that almost always chooses the optimal actions, while $\pi(a^\*|s) = 0.5$ represents a 'Random Policy'."
> >     - Then, for such policies, we can compute the probability of having keys at $L=6$, i.e., $P^\pi(X|L=6)$, which are computed by ..... The values indicate the visitation frequencies of such policies. In particular, for the "good policy", ....; for the "random policy", ....
> >     - The Q-value, defined by, ... represents...., and indicates....
> >     - The same for the advantages.
> > 3. My additional comments: "I have a more general question: do you also see such a strong policy confounding issue if you have a random starting state in training? I do not think this will remove entirely potential mislearned correlation, but it seems to me that this can help at least the naive implication of “moving to location 3 implies having a key”."
> > Again, this might sound unnecessary to you as authors. However, these are important details for the readers.
> >
> > I recommend resubmitting after some modifications.

---

> ### Author Response · Authors · 2026-01-13
>
> Thank you for the follow-up. We address each of your points below:
>
> **1. Rigor:**
>
> - *Overloaded notation:* We respectfully disagree that this notation is unclear or incorrect. In Section 4.1, $F$ is explicitly defined as a *set of random variables*, with $|F|$ denoting its cardinality, and each element $F^i \in F$ being a random variable with domain $\mathcal{F}^i$. The Cartesian product $\times \mathcal{F}$ denotes the joint domain of these variables. This convention is standard in probabilistic modeling and causal inference, and is consistent with the definitions provided in the paper.
>
> - *Definitions:* We again disagree that these definitions are not fixed. $T^\pi_\Phi$ is defined in Definition 4, $P^\pi(s_t \mid \Phi(s_t))$ is defined in Definition 8, and $P^\pi(s_t, a_t \mid \Phi(s_t))$ is defined in Theorem 1. The notation $\Pr$ is a typo and should be $P$.
>
> **2. Tables and figures:**
>
> - *Table 1:* We thank the reviewer for the detailed suggestions. The concern about mixed notation for states (Key/No Key vs. $X=0/1$) and actions (Left/Right vs. $A=0/1$) was already raised in the original review and has been addressed in the current revision. However, the additional concerns—such as providing a more explicit walkthrough of the table, formally defining $a^*(s)$, and specifying which Q-function and advantage function are reported—were not mentioned in the original review. While we appreciate this feedback and will incorporate it in the next revision, we would have preferred that these points be raised earlier so they could be addressed alongside the other clarifications.
>
> - *Figures:* We have already addressed the concerns previously raised by the reviewer regarding Figures 2 and 4. If there are additional issues or points of clarification, we would be happy to address them.
>
> **Importantly, these are presentation and exposition issues that do not affect any of the theoretical or empirical claims in the paper.**
>
> **3. Random starting state:**
>
> We apologize for not addressing this question in our first response. The reviewer is correct that if the starting state were randomized, policy confounding would be reduced, as the agent would observe a more uniform distribution of states and could not rely on spurious correlations (e.g., being at location 6 implies having a key). However, this is not a general solution: it requires explicitly modifying the environment dynamics, which is often infeasible in practice.

---

### Comment · Action_Editor_xNzb · 2025-12-16
**All reviews are submitted - discussion period starts**

Dear Authors,

All reviews have now been posted, and the discussion period is officially open.

The Reviewers raised several critical and recurring concerns that need to be addressed in your response and manuscript revision. To mention the most salient ones: there are significant clarity issues regarding definitions and notation that lead to doubts about the rigorousness of the claims; there is a lack of connections to relevant existing literature; and there are doubts over the fundamental claim of a "causal mechanism," with one reviewer suggesting the effect might simply be a better weighting scheme.

Please use this discussion period to clarify these doubts and improve the manuscript.

Looking forward to your response.

Regards,
AE

---

### Author Response · Authors · 2025-12-31
**Summary of revisions.**

We thank the reviewers for their constructive feedback. In response, we (i) clarified notation and mathematical definitions throughout, including the formal treatment of state representations, induced values, and transition dynamics; (ii) refined the theoretical analysis of the advantage function, in particular the scaling effect and its connection to policy confounding; (iii) improved the clarity and consistency of the Key2Door numerical example and accompanying table; (iv) expanded the discussion of experimental results, including the interpretation of KL divergence and out-of-trajectory generalization; and (v) strengthened the limitations section to better delineate the scope of our claims and their potential relevance to more complex domains. All substantive changes in the revised version are highlighted in blue.

---

### Decision · Action_Editor_xNzb · 2026-01-25

**Recommendation:** Reject

**Audience:**

Yes

**Audience Explanation:**

The main problem considered in the paper, policy confounding, is a significant issue in RL where agents exploit spurious correlations between observations and rewards rather than learning true causal mechanisms. While all reviewers confirmed that the topic is of interest to the TMLR audience, the evidence supporting the core theoretical claims was found insufficient. As explained above (and further detailed in the reviews), the manuscript lacks the necessary mathematical rigour and claims on causal discovery are not sufficiently supported. As a result, while relevant for the TMLR audience, the paper requires extensive re-adjustment of its theoretical part and more supporting evidence before being ready for publication.

**Claims And Evidence:**

No

**Claims Explanation:**

The authors provide compelling empirical evidence (in Key2Door, Frozen T-Maze, and Diversion) showing that advantage functions consistently outperform standard Q-learning in these settings. They also nicely demonstrate the limitations of standard PPO, showing that the common practice of mini-batch normalisation actively removes the critical probability-dependent scaling effect required to mitigate confounding.

However, the more fundamental part of the paper is considered insufficient. In particular:
1. Reviewers Cxxp and os8x remain sceptical of the formal presentation, specifically reporting missing or non-rigorous definitions for the do-operator and the projected state space.
2. Also the claims regarding causality are not sufficiently supported. Reviewer os8x argues there is a conflation between a useful "reweighting mechanism" (which effectively handles out-of-distribution samples) and true "causal discovery".

Finally, there are significant clarity issues (in particular regarding Table 1) and concerns regarding the scalability of the findings, as the experiments rely on small, illustrative environments.

**Resubmission Of Major Revision:**

The authors may consider submitting a major revision at a later time.